# Intranasal Fentanyl for Acute Pain Management in Children, Adults and Elderly Patients in the Prehospital Emergency Service and in the Emergency Department: A Systematic Review

**DOI:** 10.3390/jcm12072609

**Published:** 2023-03-30

**Authors:** Sossio Serra, Michele Domenico Spampinato, Alessandro Riccardi, Mario Guarino, Rita Pavasini, Andrea Fabbri, Fabio De Iaco

**Affiliations:** 1Emergency Department, Maurizio Bufalini Hospital, 47521 Cesena, Italy; 2Department of Translational Medicine and for Romagna, University of Ferrara, 44124 Ferrara, Italy; 3Pronto Soccorso Ospedale di Imperia, 18100 Imperia, Italy; 4UOC MEU Ospedale CTO- AORN dei Colli Napoli, 80131 Naples, Italy; 5UO Cardiologia, Azienda Ospedaliero Universitaria di Ferrara, 44124 Ferrara, Italy; 6Emergency Department, AUSL Romagna, Presidio Ospedaliero Morgagni-Pierantoni, 47121 Forlì, Italy; 7Struttura Complessa di Medicina di Emergenza Urgenza Ospedale Maria Vittoria, ASL Città di Torino, 10144 Torino, Italy

**Keywords:** intranasal administration, fentanyl, pain, emergency department, prehospital emergency service

## Abstract

This systematic review examined the efficacy and safety of intranasal fentanyl (INF) for acute pain treatment in children, adults, and the elderly in prehospital emergency services (PHES) and emergency departments (ED). ClinicalTrials.gov, LILACS, PubMed, SCOPUS, EMBASE, Google Scholar and Cochrane databases were consulted until 31 December 2022. A total of 23 studies were included: 18 in children (1 PHES, 17 ED), 5 in adults (1 PHES, 4 ED) and 1 in older people (1 PHES subgroup analysis). In children, INF was effective in both settings and as effective as the comparator drugs, with no differences in adverse events (AEs); one randomised controlled trial (RCT) showed that INF was more effective than the comparator drugs. In adults, one study demonstrated the efficacy of INF in the PHES setting, one study demonstrated the efficacy of INF in the ED setting, two RCTs showed INF to be less effective than the comparator drugs and one RCT showed INF to be as effective as the comparator, with no difference in AEs reported. In older people, one study showed effective pain relief and no AEs. In summary, INF appears to be effective and safe in children and adults in PHES and ED. More high-quality studies are needed, especially in PHES and older people.

## 1. Introduction

Pain is a common and distressing symptom in patients presenting to prehospital emergency services (PHES) and the emergency department (ED) [1,2,3,4] and prompt and appropriate treatment is crucial. The term “oligoanalgesia” was coined in 1989 to describe the undertreatment of pain in emergency departments [5,6]. According to a recent study by Todd et al., 57% of ED patients suffer from moderate/severe pain but only 50% receive adequate analgesic treatment [7]. Oligoanalgesia disproportionately affects vulnerable groups such as children [8], patients with communication difficulties [9] and the elderly [10,11]. Inadequate pain management in the ED may be due to a lack of training in pain recognition and management, opioid phobia, failure to recognise pain in vulnerable groups, underestimation of patient-reported pain and poor communication with patients [12,13,14,15].

Non-pharmacological and pharmacological interventions are used for pain management in the emergency department, with drugs administered orally, intravenously, intramuscularly (IM), intraosseously or intranasally. Oral administration is preferred but may be limited by various diseases and a slow onset of action, especially in patients with impaired consciousness or acute severe pain, making the parenteral route the preferred option [16]. The intranasal (IN) administration of drugs is a new, effective, and safe alternative to intravenous administration, especially when intravenous access is difficult, time-consuming or unnecessary. IN administration of highly lipophilic drugs, such as fentanyl, is rapid, with direct entry into the CSF and brain, avoiding the hepatic first-pass effect and making it useful for analgesia, sedation, termination of seizures, reversal of narcotics and benzodiazepines and treatment of hypoglycaemia [17,18,19,20,21,22,23].

Intranasally administered fentanyl (INF) is a well-tolerated, safe and effective method of pain management, with a bioavailability of 71–89% [24,25], therapeutic levels reached within 2 min, a time to maximum arterial concentration of 7 min [26,27] and a plasma half-life of 60 min [28]. A single dose provides analgesia lasting 120–200 min [27], with minor adverse effects limited to mild mucosal irritation [29]. IN fentanyl provides effective analgesia without the need for intravenous access or iatrogenic pain from intramuscular injections. This makes it particularly useful for patients with minor injuries who do not require intravenous access for resuscitation [30,31]. Holdgate et al. found that the use of IN fentanyl significantly reduced the time from patient arrival to initial analgesia compared with intravenous morphine [32]. Although most of the published studies have been conducted in the paediatric population using the standard fentanyl solution of 50 μg/mL, few studies have investigated the use of intranasal fentanyl in adults and elderly patients, mostly using fentanyl concentrations of 300–1000 μg/mL [33,34,35], suggesting that INF is a promising option for safe and effective pain management in both PHES and ED.

The aim of this systematic review was to evaluate the efficacy and safety of intranasally administered fentanyl at a concentration of 50 µg/mL in children, adults, and elderly patients with acute pain in prehospital emergency services (PHES) and in the ED.

## 2. Materials and Methods

The methodology and reporting of this systematic review were in accordance with the 2021 version of the preferred reporting items for systematic reviews and meta-analyses (PRISMA) statement [36]. This systematic review was registered in the Open Science Framework on 26 February 2023. (Registration DOI: https://doi.org/10.17605/OSF.IO/PHMC8) 

### 2.1. Search Methods and Data Extraction

This systematic review was conducted on 31 December 2022. The following databases were searched: MEDLINE, SCOPUS, EMBASE, SCHOLAR, “Cochrane Central Register of Controlled Trials” (CENTRAL), “Latin American and Caribbean Health Science Information Database” (LILACS) and ClinicalTrials.gov. The following terms were searched using the medical subject heading (MeSH) strategy: “intranasal” OR “intranasally” AND “fentanyl” AND ((“emergency service, hospital” OR “hospital emergency service” OR (“emergency” AND “department”) OR “emergency department”) OR (“acute pain” OR (“acute” AND “pain”) OR “acute pain”)). The search strategy was modelled for each database. The search string for each database was agreed upon by all authors. Each database was screened by two independent reviewers (SS and SMD) and the titles and abstracts were checked for relevance. If no abstract was available, the full text was analysed for inclusion. The reference lists of review articles and relevant studies, textbooks and abstracts were also reviewed to include potentially relevant articles. Disagreements about eligibility were resolved by discussion and referrals to a third reviewer (FDI). Two independent reviewers (AR and MG) extracted and entered the data into an electronic data sheet. 

The following data were extracted: (i) general characteristics of included studies (author, year of publication, type of study (randomised clinical trial (RCT), prospective study (PS) or retrospective study (RS)), sample size, age of included patients, pain scale used, INF dose administered, type and dose of comparator drug (if available), population, primary outcomes and author’s conclusion); (ii) efficacy of INF (cause of pain, pain at baseline, pain at any time point evaluated for INF and comparator and other efficacy outcomes assessed by each study); (iii) safety of INF (respiratory depression for INF and comparator, cardiovascular depression for INF and comparator, CNS depression for INF and comparator and other adverse events assessed for INF and comparator). Only the data available in the original manuscript were used for this systematic review and no author was contacted for further data or specifications. All data were reported as in the original study.

### 2.2. Types of Studies Included

All randomised clinical trials, prospective observational studies and retrospective studies evaluating the effect of 50 μg/mL fentanyl IN delivered for acute moderate to severe pain due to acute traumatic injury (e.g., fractures, burns and wounds, confirmed or suspected) or acute medical illness in the setting of PHES or ED without a comparison group or compared with (i) administration of other pharmacological interventions for pain control, (ii) non-pharmacological interventions and (iii) placebo administration were included. Systematic reviews, narrative reviews, studies published in languages other than English, case reports, guidelines, surveys, study protocols, INF administered in fixed combination with other drugs, studies on procedural analgesia, other settings, other IN drugs, non-human populations, non-available full-text studies, and studies evaluating >50 μg/mL fentanyl concentration were excluded.

Studies were included regardless of the device used for IN administration (droplet, atomizer, or spray).

Studies that evaluated IN fentanyl (i) as part of procedural sedation and analgesia (i.e., to make painful procedures more tolerable), (ii) for perioperative pain, (iii) cancer-related pain including breakthrough pain, (iv) chronic pain or other non-acute pain, (v) in patients on chronic opioid therapy and (vi) given in combination with other medications were excluded. 

### 2.3. Evaluated Outcomes

The primary outcomes were: (1) the efficacy of INF reported as a reduction in pain score assessed via appropriate scales (Wong–Baker, faces, legs, activity, cry, consolability (FLACC) scale, numeric rating scale (NRS), visual analogic scale (VAS), pain assessment in advanced dementia (PAINAD)) or quality assessment in terms of no pain, mild, moderate or severe and the type of scale used was reported. The efficacy of INF was also reported according to each study, such as the need for rescue analgesia, time to opioid administration, percentage of “relevant analgesia” (the exact definition used in each study was also provided if available), admission rate or patient or parent satisfaction; (2) the safety of INF was reported as follows: any respiratory, cardiovascular (CV) or central nervous system (CNS) depression was reported and defined as “minor”, if no abnormal vital signs were reported at any time, or as “major”, if abnormal vital signs were reported (respiratory rate < normal range for age, peripheral oxygen saturation <92% or need for oxygen administration or mechanical ventilation, hypotension or bradycardia according to normal value for age or need for vasopressors or inotropic agents, a Glasgow coma scale (GCS) <14 or defined “moderately sedated” according to the sedation scale used), or death. Other minor adverse effects attributable to fentanyl administration, such as nausea, vomiting, dizziness and drowsiness, or attributable to intranasal administration, such as bad taste, itchy nose and unpleasant taste, have also been reported.

### 2.4. Quality Assessment of Included Studies

Risk of bias for the RCTs was assessed using the Cochrane risk of bias 2.0 tool for the outcome of pain reduction. A judgement (low risk of bias (L), high risk (H) or some concern (SC)) was assigned for each of the following domains: (i) risk of bias arising from the randomization process, (ii) risk of bias due to deviations from the intended interventions (both effect of assignment to intervention and effect of adhering to intervention), (iii) risk of bias due to missing outcome, (iv) risk of bias in the measurement of the outcome, (v) risk of bias in the selection of the reported result and (vi) overall risk of bias.

The quality of observational studies was assessed using the methodological index for non-randomized studies (MINORS) criteria [37]. 

The risk of bias of RCTs and the quality of observational studies were assessed independently by two review authors (SS and SMD). Disagreements were resolved in a consensus discussion involving a third author (FDI).

## 3. Results

### 3.1. Result of the Database Research

A total of 911 studies were identified (153 in MEDLINE, 159 in SCOPUS, 347 in EMBASE, 232 in SCHOLAR, 96 in CENTRAL, 3 in LILACS and 6 in ClinicalTrials.gov). 

After controlling for titles and abstracts, 141 studies were removed. After controlling for duplicates (N = 114), 449 studies were analysed, of which 23 were included in the present systematic review (see Figure 1 for the PRISMA flow diagram and Table 1, Table 2 and Table 3 for the general characteristics of the included studies). 

### 3.2. General Characteristics of the Included Studies

Out of 23 studies included, 18 studies were conducted in children: 1 PS in the PHES (Murphy et al., 2017 [38]), 9 observational studies (Akinsola et al., 2018 [39], Anderson et al., 2022 [40], Cole et al., 2009 [41], Crelin et al., 2010 [42], Finn et al., 2010 [43], Kelly et al., 2018 [44], Nemeth et al., 2019 [45], Saunders et al., 2010 [46], Schaefer et al., 2015 [47]) and 8 RCTs (Borland et al., 2011 [48], Fein et al., 2017 [49], Frey et al., 2019 [50], Graudins et al., 2015 [51], Quinn et al., 2021 [52], Reynolds et al., 2017 [53], Ruffin et al., 2022 [54], Younge et al., 1999 [30]) in the ED; 5 studies were conducted in the adult population: 1 observational study in the PHES setting (Tanguay et al., 2020 [55]), 2 observational studies (Assad et al., 2023 [56], Belkouch et al., 2015 [57]) and 2 RCTs (Nasr Isfahani et al., 2022 [58], Nazemian et al., 2020 [59]) in the ED setting; one subgroup analysis of an observational study reported data on the elderly patients in the PHES (Tanguay et al., 2020 [55]). A total of 10,280 patients were included: 1203 in the PHES setting (94 children, 729 adults and 380 elderly) and 9077 in the ED setting (8714 children and 363 adults). The population, the pain scale used, the INF dose, the comparators, the primary outcomes and the authors’ conclusion for each included study are shown in Table 1, Table 2 and Table 3. 

**Table 1 jcm-12-02609-t001:** General characteristics of the included studies conducted on children.

Author, Year of Publication	Type of Study *	Sample Size N (N of Women Included, (%)), N in INF Group/N in the Comparator Group (if Available)	Age ^£,^**	Pain Scale Used	INF Dose **	Comparator	Population, Primary Outcomes and Authors’ Conclusions
Prehospital emergency service setting							
Murphy et al., 2017 [38]	P	94 (44 (47))	11 (7–13)	FLACC or the Wong–Baker faces or the VNR according to age.	1.5 μg/kg	INF+additional analgesia ^$^	In children aged 1–16 y-o, INF at a dose of 1.5 µg/kg appears to be a safe and effective analgesic in the prehospital management of acute severe pain.
Emergency department setting							
Akinsola et al., 2018 [39]	P	228 (128, (56))–180/48	12 ± 5	NR	1.5 μg/kg, two doses 5 min apart	Standard care (± oral hydrocodone, ± IV ketorolac and ± IV morphine or IV hydromorphone)	In children with pain due to vaso-occlusive crisis, INF use significantly improved time to first parenteral-opioid dose and was a safe and effective treatment for pain.
Anderson et al., 2022 [40]	R	3205 (1263, (40))	13.7, (11.8–15.9)	NR	2–5 μg/kg, maximum 200 μg	NC	In children, higher doses of fentanyl (2–5 μg/kg) are well tolerated without any clinically significant adverse outcomes observed over a 7-year period.
Borland et al., 2011 [48]	RCT	189 (118 (63))	9.1 (95% CI 8.4–9.8) for INF, 8.8 (95% CI 8.1–9.5) for comparator	VAS or FPS-R	1.5 μg/kg	1.5 μg/kg of INF delivered with a concentration of 300 μg/mL	In children aged 3–15 y-o with pain due to suspected long bone fracture, standard concentration fentanyl and high concentration fentanyl were equivalent in reducing pain.
Cole et al., 2009 [41]	P	46, (24, (52))	22.6 (12–36) months	FLACC	1.5 μg/kg, a second dose of 0.5 μg/kg if persistent pain after 10 min	NC	In children aged 1–3 years with acute moderate to severe pain, INF is an effective, safe and well-tolerated mode of analgesia.
Crellin et al., 2010 [42]	P	36	6.7, range 5–15	VAS or Bieri faces scale-revised	1.5 μg/kg	NC	In children aged 5–18 y-o with upper limb injuries, INF appeared to be an effective analgesic.
Fein et al., 2017 [49]	RCT	49 (19 (39))–24/25	10.6 (5.3) for INF, 12.5 (5.1) for comparator	Modified Wong–Baker faces pain rating scale	2 μg/kg (maximum 100 μg), single dose	SoC and IN 0.9% NaCl	In children aged 3–20 y-o with a vaso-occlusive crisis and pain score > 6 at WBFPRS, at 20 min, INF reduced vaso-occlusive crisis pain more than placebo.
Finn et al., 2010 [43]	R	49 (0 (0))	6.2, range 1–16	VAS	1.5 μg/kg	NC	In children aged 1–16 y-o, this study shows INF to be both effective and safe.
Frey et al., 2019 [50]	RCT	90 (29 (32))–45/45	12.2 (2.3) for the INF group; 11.8 (2.6) for the comparator	VAS	2 μg/kg	1.5 mg/kg IN ketamine	IN ketamine provides effective analgesia that is non-inferior to INF, although participants who received IN ketamine had an increase in adverse events that were minor and transient.
Graudins et al., 2015 [51]	RCT	73 (27 (37))–37/36	9 (6–11) for INF, 7 (6–9.5) for comparator	VAS	1.5 μg/kg	1 mg/kg IN ketamine	In children aged 3–13 y-o with isolated limb injury and pain at least 6/10 at triage, INF and IN ketamine were associated with similar pain reduction. IN Ketamine was associated with more minor adverse events.
Kelly et al., 2018 [44]	R	487 (170 (35))–376/111	In INF group: 16.3 ± 4.8; in comparator group: 18.2 ± 3.6	NRS	NR	Routine care, drugs and dosage NR	In children aged 1–21 y-o with acute pain due to vaso-occlusive events compared with routine care, INF demonstrated a significantly reduced time to initiation of opioid analgesic therapy when using INF.
Nemeth et al., 2019 [45]	P	100 (42 (42))–19/7/5/1/1/1	5.5 ± 4.1	FLACC, faces pain scale revised or NRS according to age	2.0 μg/kg	S-ketamine, midazolam via IV, PO or PR in various combinations	In children aged 0–17 y-o with trauma for analgesia or procedural sedation, intranasal administration of fentanyl, s-ketamine and midazolam was shown to be generally rapid for achieving analgesia and/or sedation. No marked circulatory, respiratory or other SAEs were noted.
Quinn et al., 2021 [52]	RCT	22 (4 (18))–11/11	INF group: 9.58 ± 2.92; comparator group: 9.77 ± 2.51; *p* = 0.87	NRS or Wong–Baker faces pain score	1.5 μg/kg	1 mg/kg IN ketamine	In children aged 3–17 y-o, IN ketamine was found to be inferior to IN fentanyl in relieving pain at 10 min and was found to have significantly greater rates of sedation and dizziness. No sufficient power to support the non-inferiority of IN ketamine compared with INF at 20 min after administration.
Reynolds et al., 2017 [53]	RCT	82 (31 (38))–41/41	8 (5–11)	NRS or Wong–Baker faces pain scale	1.5 μg/kg	1 mg/kg IN ketamine	In children 4–17 y-o with acute pain from suspected isolated extremity fractures with pain score >3 on the Wong–Baker faces pain scale or >2 at NRS, IN ketamine was associated with more minor side effects than intranasal fentanyl. Pain relief at 20 min was similar between groups.
Ruffin et al., 2022 [54]	RCT	34 (17 (50)), 17/17	INF group: 3.1 years; comparator group: 1.8 years, *p* = 0.06	Faces, FLACC or VAS according to age	1.5 μg/kg	PO administered acetaminophen+hydrocodone, 0.15 mg/kg hydrocodone	In 6 months–18 y-o children with painful infectious mouth conditions, INF seems to be a safe and effective alternative to acetaminophen with hydrocodone in reducing pain.
Saunders et al., 2010 [46]	P	81 (32 (39))	8 ± 3.7	Wong–Baker faces scale or VAS according to age	2 μg/kg	NC	In children aged 3–18 y-o with moderate to severe pain on the Wong–Baker faces scale or VAS, a single dose of INF provides effective analgesia for paediatric ED patients with painful orthopaedic trauma within 10 min of administration.
Schaefer et al., 2015 [47]	R	54 (36 (66))–7/47	INF group: 7.7 ± 4.7 comparator group: 13.4 ± 3.8, *p* = 0.018	NRS or faces pain score	1.1 to 1.5 μg/kg	IV opioids administration	INF administration reduces the time from physician encounter to opioid administration in paediatric patients.
Younge et al., 1999 [30]	RCT	47 (30 (63))–24/23	INF group: mean 6.6 (SD NR) comparator: 7.1 mean (SD NR), *p* = 0.053	NR	1 μg/kg	0.2 mg/kg IM morphine	In children aged 3–10 y-o, INF provides effective pain relief for children requiring opioid analgesia in the ED. It appears as effective, with better tolerance to administration, as IMM

Note: *: RCT—randomised controlled trial, P—prospective cohort study, R—retrospective cohort study; ** expressed as media±standard deviation (SD) or median (IQR); ^£^ in years unless otherwise specified; ^$^ additional analgesia: ± paracetamol ± ibuprofen ± inhaled nitrous oxide; ED—emergency department; FLACC—face, leg, activity, cry, consolability; FPS-R—face, pain scale revised; IM—intramuscular; IMM—intramuscular morphine; IV—intravenous; IN—intranasal; INF—intranasal fentanyl; NR—non-reported; NC—no comparator; PO—orally administered; PR—rectal administered; SoC—standard of care; VAS—visual analogue scale; y-o—years old; WBS—Wong–Baker faces scale.

**Table 2 jcm-12-02609-t002:** General characteristics of the studies conducted in the adult population.

Author, Year of Publication	Type of Study *	Sample Size N (N of Women Included, (%)), N in INF Group/N in the Comparator Group (if Available)	Age in Years **	Pain Scale Used	INF Dose	Comparator	Population, Primary Outcomes and Authors’ Conclusions
Prehospital emergency service setting							
Tanguay et al., 2020 [55]	R (subgroup analysis for patients aged 18–70 y-o)	729–402/327	59 ± 19.9	NRS	1.5 μg/kg, maximum dose of 100 μg, 50 μg in patients with general comorbidities ^£:^	1.5 μg/kg SC fentanyl	In patients aged 18–70 y-o, both INF and SCF are feasible, effective and safe for managing acute severe pain in the prehospital setting. We also found that a greater proportion of older patients in the INF group experienced pain relief, even though they received a lower dose of fentanyl.
Emergency department setting							
Assad et al., 2023 [56]	R	95–31/64	31.1 (10.4) for INF, 31.8 (9.2) for comparator, *p* = 0.5	NR	50 μg or 100 μg	IV morphine, 0.1 mg/kg	INF provided similar pain reduction compared to IV morphine in the treatment of adults with VOC presenting to the ED; however, there was a trend in readmission within 48 h. No significant difference in adverse events between the groups.
Belkouch et al., 2015 [57]	P	23 (11 (47.8))	51.3	VAS	1.5 µg/kg	NC	In patients admitted for renal colic, INF provides quick pain relief and its use is safe.
Nasr Isfahani et al., 2022 [58]	RCT	125 (9 (8)) –44/40 in IN ketamine/41 in IN placebo	INF group: 30.51 ± 10.77; for placebo group: 32.25 ± 13.23; IN ketamine group: 31.26 ± 12.07 (*p* > 0.05)	VAS	1 µg/kg	1 mg/kg IN ketamine and IN placebo	In patients with isolated limb trauma aged 15–65 y-o with moderate to severe pain (at least 45 mm at VAS), the efficacy of INF and IN ketamine was similar in reduction of pain 40 min after the administration. IN ketamine has a reduced time of onset. The rate of minor adverse events after IN l ketamine was higher than INF without serious adverse events registered.
Nazemian et al., 2020 [59]	RCT	220 (96 (43)) –110/110	NR	NRS	2 µg/kg + 60 mg IM ketorolac	1 µg/kg IV fentanyl + 60 mg IM ketorolac	In patients with renal colic pain, INF in combination with ketorolac is a fast-acting, non-invasive, convenient and effective method to manage pain in these patients.

Note: * RCT—randomised controlled trial; P—prospective cohort study; R—retrospective cohort study; ** expressed as media ± SD or median (IQR); ^£^—defined as chronic obstructive pulmonary disease, general weakness or malnutrition; IM—intramuscular; IN—intranasal; INF—intranasal fentanyl; IV—intravenous; NC—no comparator; NR—non-reported; NRS—numeric rating scale; VAS—visual analogue scale; SC—subcutaneous; y-o—years old.

**Table 3 jcm-12-02609-t003:** General characteristics of the studies included in the elderly population.

Author, Year of Publication	Type of Study *	Sample Size N (N of Women Included, (%)), N in INF Group/N in Comparator Group (if Available)	Age in Years **	Pain Scale Used	INF Dose	Comparator	Population, Primary Outcomes and Authors’ Conclusions
Prehospital emergency service setting							
Tanguay et al., 2020 [55]	R (subgroup analysis for patients aged >70 y-o)	380–202/195	NR	NRS	50 μg	50 μg SC fentanyl	In patients aged >70 y-o with severe pain, both INF and SCF are feasible, effective and safe for managing acute severe pain in the prehospital setting. We also found that a greater proportion of older patients in the INF group experienced pain relief, even though they received a lower dose of fentanyl.

Note: * RCT—randomised controlled trial; P—prospective cohort study; R—retrospective cohort study; ** expressed as media ± SD or median (IQR); INF—intranasal fentanyl; NRS—numeric rating scale; SC—subcutaneous; SCF—subcutaneous fentanyl; y-o—years old.

### 3.3. Efficacy of Intranasal Fentanyl

#### 3.3.1. Efficacy in Children, PHES Setting

One study reported efficacy in reducing pain at 10 min after INF, without differences reported with traditional analgesia [38].

#### 3.3.2. Efficacy in Children, ED Setting

Four studies reported INF to be effective in reducing pain at different time points (Cole et al., 2009 at 10 and 30 min [41]; Crellin et al., 2010 [42]; Finn et al., [43] at 5 and 30 min; Saunders et al., 10, 20 and 30 min [46]). Two studies [44,45] and six RCTs reported INF to be equally effective as comparators [48,50,51,52,53,54]. Two RCTs demonstrated INF to be more effective in reducing pain: Fein et al., 2017, compared INF to the “standard of care” (more effective at 20 min after administration, no differences at 10 and 30 min) [49] and Younge et al., 1999, compared IM 0.2 mg/kg morphine (more effective at 10 min, no difference at 20 and 30 min) [30]. Several authors have reported different secondary efficacy outcomes: one RCT reported an increased need for additional analgesia in patients treated with 50 μg/mL INF compared with 300 μg/mL INF [48], while other RCTs reported no difference between INF and 1 mg/kg IN ketamine [51,52] and 1.5 mg/kg IN ketamine [50]; one PS [45] reported no difference from comparators. Three studies reported a reduction in the time to first opioid administration in patients treated with INF [39,44,47] and an increased percentage of pain relief at 30 min after ED arrival [44]. One PS also reported a reduced ED length of stay and admission rate in the INF group [39]. One RCT reported no change in overall satisfaction or pain score at 60 min [54] and one RCT reported higher tolerance to INF compared with IM morphine [30].

#### 3.3.3. Efficacy in Adults, PHES Setting

One RS reported a higher proportion of patients with clinically significant pain relief in patients treated with INF than with subcutaneous fentanyl, with no differences in time to drug administration [55]. 

#### 3.3.4. Efficacy in Adults, ED Setting

One PS reported that INF effectively reduced pain at each time point after administration (5, 30, 45 and 60 min) [57]; one RS reported no difference in pain reduction with IV morphine [56]; one RCT reported a reduced efficacy at 5 min and 10 min compared with IN ketamine, with no difference at 30 and 40 min [58]; one RCT reported a reduced efficacy of INF compared with IV fentanyl at each time point after administration [59]. Regarding secondary efficacy outcomes, one RS reported no difference in time to first analgesic administration between INF and IV morphine. However, INF resulted in higher milligrams of morphine equivalents and a lower percentage of patients discharged home from the ED [56]; one RCT reported lower rescue analgesia with INF than IN ketamine and higher satisfaction levels [58]; one RCT reported no difference in satisfaction levels between INF and IV fentanyl [59].

#### 3.3.5. Efficacy in the Elderly, PHES Setting

One RS reported a greater INF analgesic effect among patients aged >70 years, but precise numerical pain data for INF and the comparator were not reported. In this study, patients aged >70 years received an INF dose of 50 µg, with a rescue dose of 25 µg after 15 min in case of ineffective analgesia. The time of fentanyl administration was similar for both routes of administration (INF, mean 9 min 36 s (s) (SD 3 min 32 s); SCF, mean 9 min 30 s (SD 3 min 46 s), *p*-value 0.674) [55].

#### 3.3.6. Efficacy in the Elderly, ED Setting

No data were reported.

All data regarding the efficacy of INF according to each included study are reported in Appendix A.

### 3.4. Safety of Intranasal Fentanyl

#### 3.4.1. Safety in Children, PHES Setting

One PS reported no respiratory, CV or CNS depression or secondary adverse events [38]. 

#### 3.4.2. Safety in Children, ED Setting

Among all included studies, adverse respiratory, CV or CNS events were reported in five studies. One RCT reported sedation in 16% and 24% of patients treated with INF and INF at higher concentrations (*p* = 0.18) [48]; one RCT reported transient hypoxia in 13% of patients, transient hypotension in 8% and drowsiness in 42% of patients (no further specification), with no difference reported with the comparator drugs and all events requiring no intervention [49]; one PS reported an UMSS score = 1 in 21% of patients and n UMSS = 2 in 5% of patients, without differences with the comparator [45]; one RCT reported a significantly lower rate of sedation due to INF than IN ketamine (0 vs. 64%, *p* = 0.004) [52]; one RCT reported no difference in sedation and 2% of patients with transient mild hypotension without requiring interventions [53]. Secondary minor adverse effects attributable to fentanyl or the IN route were also reported in 11 studies: dizziness was reported in 1% [48], 9% [52], 15% [53] and 17% of patients [51]; nausea in 4% [48], 8% [49] and 7% of patients [53]; vomiting in 1% of patients [48], nasal burning in 13% of patients [48], itchy nose in 12% of patients [51], unpleasant taste in two patients [50] and 22% of patients [53]. Although fewer minor adverse events occurred with INF than with IN ketamine [52,53], no differences in the frequency of adverse events due to the route of administration were reported.

#### 3.4.3. Safety in Adults, PHES Setting

One RS did not report adverse events [55].

#### 3.4.4. Safety in Adults, ED Setting

One RS reported bradycardia within 6 h of administration in 12.9% of patients, with no difference from IV morphine or need for intervention [55]; one RCT reported sedation in 4.5% and 2.3% of patients after 15 and 30 min from administration, respectively, with no difference from IN ketamine [58]; minor adverse events were reported by Nasr Isfahani et al., 2022 [58] (general discomfort in 6.8% of patients with no difference from IN ketamine) and by Nazemian et al., 2020 [59] (nausea in 8.2%, dizziness in 2.7%, itching in 2.7%, bad taste in 10.9%, throat irritation in 8.2%, with no difference from IV fentanyl).

#### 3.4.5. Safety in Elderly, PHES Setting

One RS did not report adverse effects in this population [55].

#### 3.4.6. Safety in the Elderly; ED Setting

No study available.

All data regarding the safety of INF according to each included study are reported in Appendix B.

### 3.5. Risk of Bias for the Outcome of Efficacy in Reducing Pain in the Included RCTs

According to the Cochrane R.o.B. 2.0 tool, in children, six of the eight RCTs conducted in the ED setting were rated as low risk of bias [48,49,50,51,52,53], while two RCTs were rated as high risk of bias [30,54]. 

In adults, of the two RCTs conducted in the ED setting, one RCT was rated as low risk [58] and one RCT was rated as high risk of bias [59] (Table 4).

According to the MINORS criteria, in children, one study in the PHES setting was rated 13 out of 16 points; five studies in the ED setting were rated 10 to 16 out of 16 points and four studies were rated 14 to 17 out of 24 points. In adults, one study conducted in the PHES setting was rated 20 out of 24 points and two studies conducted in the ED setting were rated 14 to 19 out of 24 points (Table 5).

## 4. Discussion

The aim of this systematic review was to assess the evidence for intranasal administration of fentanyl for the relief of acute pain of any cause in the emergency setting, both in the prehospital and emergency department.

This systematic review focused on the standard concentration of 50 µg/mL and excluded studies that investigated the use of higher concentrations of fentanyl, as these are not routinely available and may result in different absorption rates due to the different volumes administered into each nostril. In addition, due to the complex pathophysiology of chronic pain and cancer-related pain, studies that investigated INF for the treatment of breakthrough pain were excluded from this systematic review.

A total of 23 studies have shown INF to be safe and effective in children in both the PHES setting [38] and the ED setting [39,42,45,46], including children aged 1–3 years [40] and at higher doses administered [39], with no difference in efficacy between the standard 50 µg/mL concentration and the higher concentration [48]. Furthermore, INF is as effective as orally administered paracetamol and hydrocodone [54], as effective with a lower rate of adverse events and discomfort than IN ketamine [50,51,52,53] and IM morphine [30] and more effective than standard treatment plus IN placebo at 20 min after drug administration [48], improving the time to opioid administration [39,44,47]. In adults, one study in the PHES setting showed INF to be effective, feasible and safe, particularly in older people. Four studies have shown INF to be rapid and effective in the ED setting [57], equivalent to IV morphine [56] and IN ketamine [58], as effective as IV fentanyl at 30 min but less effective at 10, 20 and 60 min [59]. Despite the very few minor adverse events reported in the included studies, older adults with obstructive sleep apnoea, chronic obstructive pulmonary disease, heart disease and diabetes mellitus may be at a higher risk for adverse respiratory effects [60,61] and further studies focusing on this population are needed.

The included studies were heterogeneous in terms of methodology, cause of pain, scale used for pain assessment, comparison groups, INF dosing and secondary efficacy endpoints studied and deserve to be highlighted (Table 1, Table 2 and Table 3).

Of the 23 studies, we included only 10 RCTs in the ED, of which 8 were in children and 2 in adults. Two RCTs in children [30,54] and one in adults [59] were assessed as having a high risk of bias due to the unblinded methodology of the studies, which may have influenced the results. No RCTs conducted in the context of PHES or in older patients were included in this systematic review, highlighting the need for additional high-quality studies in this setting and population. However, in 2007, Rickard et al. published the results of a RCT that compared the efficacy and safety of 180 µg INF, administered via a fentanyl concentration of 300 µg/mL, with that of 2.5–5 mg IV morphine. According to the results of this study, INF and IV morphine demonstrated no difference in effectiveness, safety, and acceptability [34]. 

The included studies investigated the efficacy of INF for the medical, traumatic, and combined causes of pain. As already known, pain is defined as “an unpleasant sensory and emotional experience that is associated with or resembles actual or potential tissue damage” [62], pointing out how psychological and emotional distress could affect pain experience. Seven of the included studies investigated the use of INF for medical reasons: in children, three out of four studies were conducted for pain caused by vaso-occlusive crises due to sickle cell disease; in adults, two out of three studies were conducted for pain due to renal colic. Both conditions cause very severe, persistent, and frequently recurring pain that lasts for a prolonged period and causes a level of distress that can greatly affect the perception of pain [63,64,65]. Similarly, traumatic pain is often associated with great psychological stress due to the traumatic event and the fear of anatomical dysfunction. As is well known, IN is an unusual route of drug administration and the route of administration leads to different placebo or nocebo effects, potentially strongly influencing pain perception [66,67,68]. Of the 23 included studies, only 7 compared the administration of INF with other drugs or placebo IN, so the confounding effect of this route of administration could not be assessed. Additionally, pain assessment modalities varied widely across the included studies. The numerical rating scale (NRS), verbal rating scale (VRS) and visual analogue scale (VAS) are the most used and recommended scales for assessing pain intensity [69]. Several different scales were used in the studies included in this systematic review, including the NRS, VRS, VAS and several others, particularly in studies with children, where multiple scales were used in a single study. These included the modified Wong–Baker faces pain rating scale (WBFPRS), the Bieri faces scale and the FLACC (face, leg, activity, cry, consolability) scale. Despite the large differences between these scales, only Saunders et al., 2010 [46] reported the effectiveness of INF depending on the scale used in the subgroup analysis. However, different scales can lead to different pain intensity ratings. Pain catastrophising has a major impact on pain intensity [70] and verbal rating scales of pain severity may also reflect patients’ perceptions of pain disorders and beliefs about their pain [71]. As previously reported, the lack of detail on pain intensity ratings leads to ambiguity in the interpretation of research findings [72]. In addition, none of the included studies used the pain assessment in advanced dementia scale (PAINAD) and the results might be different in frail older patients with neurodegenerative diseases.

The studies were also very heterogeneous in terms of INF dosing, making it difficult to draw firm conclusions about the best dose in terms of efficacy and safety. Most of the included studies looked at 1.5 μg/kg INF and showed no difference in pain relief compared with INF and other drugs, IN ketamine or oral paracetamol plus hydrocodone. Of the studies examining 2 μg/kg INF, three reported no difference from standard treatment, one reported greater efficacy at 20 min than standard treatment [48] and one reported less efficacy than IV fentanyl at any time except 30 min after administration, with fewer adverse effects (without reaching statistical significance). One study investigated 1 μg/kg INF and showed lower efficacy at 5 and 10 min compared with IN ketamine, with no difference at subsequent time points.

### 4.1. Implications for Clinical Practice

The IN route proved to be a safe and rapid method of drug administration, with only minor side effects such as transient nasal itching, nasal burning, and cough. IN-administered fentanyl proved to be effective and well tolerated, with no serious adverse events. In clinical practice, INF can be used in paediatric and adult patients in prehospital emergency care and in the emergency department for acute pain of both medical and traumatic origin, especially in cases where intravenous access is required for pain management only. The results of this systematic review are consistent with those of previous SRs. In 2012, Hanses et al. published a systematic review that included 16 RCTs on the use of INF for acute pain. According to their results, there were no significant analgesic differences between IN fentanyl and IV morphine, oral morphine or IV fentanyl in the treatment of acute pain after long bone fractures, in burn patients or in the relief of postoperative pain and a significant analgesic effect of IN fentanyl was demonstrated in the treatment of breakthrough pain in cancer patients. However, no data on the safety of this route of administration have been presented [73]. Murphy et al. conducted a systematic review of the use of INF in children in 2014, which included only three studies conducted in children over three years of age with traumatic causes of pain. Although no firm conclusions could be drawn regarding the superiority, equal efficacy, or inferiority of INF over IV or IM morphine, INF was found to be an effective analgesic treatment in patients with acute moderate or severe pain, with minimal distress and no adverse events [74]. In 2018, Setlur et al. conducted a systematic review of the use of INF in paediatric patients, which included four observational studies and six RCTs. They concluded that INF was effective in relieving pain, with no differences from comparator drugs; only one RCT showed its superiority over IM morphine, again with no serious adverse events, with mild adverse effects in 3.3–3.9% of cases [75]. Abebe et al., 2021, conducted a systematic review to assess the preferred medications for pain relief in paediatric patients before hospitalisation. This showed that IN fentanyl (as well as inhaled methoxyflurane) appears to be the preferred drug for prehospital analgesia, is easy to administer, has a rapid onset and short duration of action and is as effective as morphine [76]. The PHES setting is challenging, and more studies should be conducted in this specific setting. According to the “Italian Intersociety Recommendations for pain management in emergency settings” [77], an ideal prehospital analgesic should be easy to use, safe and effective and have a predictable dose–response relationship with rapid onset and a short duration of action; many of these properties are achieved by INF. In 2014, Karlsen et al. published the results of a prospective observational study investigating the safety of INF in the PHES setting. They included 903 patients, who received a single dose of 50 μg INF, aged > 65 years and aged 18–65 years with general diseases (chronic obstructive pulmonary disease, general weakness, or malnutrition) or 100 μg INF in adults without comorbidities, using a fentanyl concentration of 500 μg/mL or 1000 μg/mL. According to their results, INF demonstrated mild adverse events in 39 patients (4.3%), with transient hypotension in 14 (1.6%), a transient drop in GCS to 14 in 5 patients (0.6%), nausea in 6 patients (0.7%), and no serious or life-threatening adverse events. No adverse events related to age were reported and similar pain relief was observed in patients with different INF dosages [33].

### 4.2. Limitations of the Present Systematic Review

This systematic review included all studies reporting the administration of fentanyl at the standard concentration of 50 μg/mL via IN for the treatment of acute pain. However, some limitations of the inclusion criteria must be discussed. Despite the broad inclusion criteria, only a few double-blind RCTs were included; more rigorous high-quality RCTs are needed. Second, we included RCTs and observational studies without sufficient blinding, which increases the risk of over- or underestimating efficacy, which is difficult to assess and could potentially lead to inaccurate conclusions. Third, the heterogeneity of the studies prevented meta-analysis and only a narrative analysis of the studies was reported. In addition, no sensitivity analysis or meta-regression was performed to evaluate the heterogeneity. Fourth, studies not in English were excluded, potentially altering the conclusions of this systematic review. Fifth, the authors were not contacted to obtain the study protocols or full study information. Sixth, we excluded studies conducted in patients with chronic pain, breakthrough pain, postoperative pain and procedural pain, therefore the conclusions of this SR may not be applicable in these settings.

## 5. Conclusions

Intranasally administered fentanyl is effective and safe for the relief of acute medical and traumatic pain, particularly in children and in the ED. INF was found to be at least as effective as IV and IM morphine, IV fentanyl and IN ketamine, with fewer adverse events than the comparator drugs. INF demonstrated efficacy and safety in children in the PHES setting, in the ED setting and in adults. However, this systematic review highlights the need for additional high-quality studies conducted in both PHES and elderly patients.

## Figures and Tables

**Figure 1 jcm-12-02609-f001:**
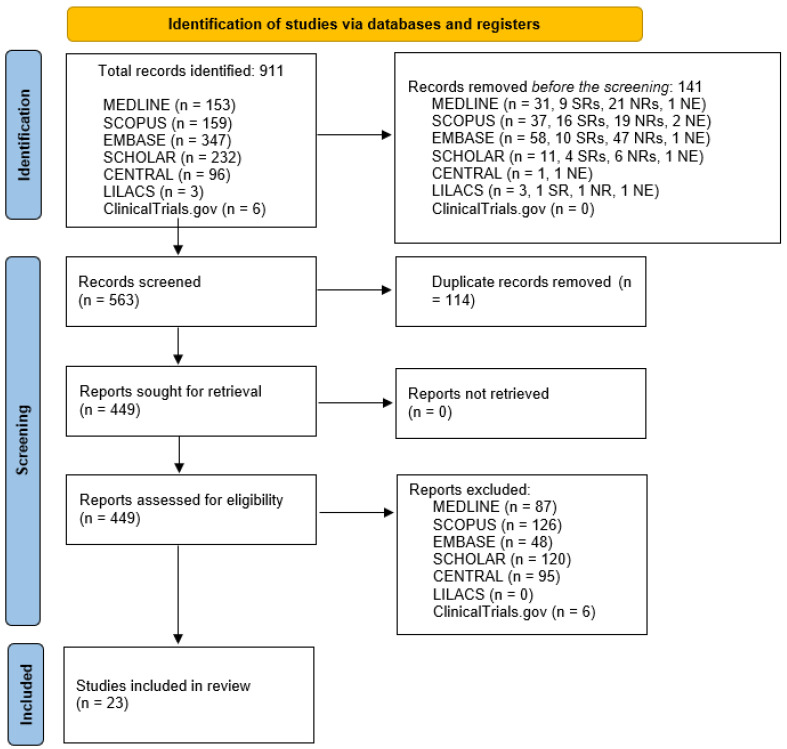
PRISMA 2020 flow diagram for identification, screening and study inclusion.

**Table 4 jcm-12-02609-t004:** Risk of bias assessment using Cochrane’s risk of bias 2.0 criteria for RCTs included.

Author, Year of Publication	Risk of Bias Arising from the Randomisation Process	Risk of Bias Due to Deviations from the Intended Interventions (Effect of Assignment to Intervention)	Risk of Bias Due to Deviations from the Intended Interventions (Effect of Adhering to Intervention)	Risk of Bias Due to Missing Outcome Data	Risk of Bias in the Measurement of the Outcome	Risk of Bias in Selection of the Reported Result	Overall Risk of Bias
Children, emergency department setting:							
Borland et al., 2011 [48]	L	L	L	L	L	L	L
Fein et al., 2016 [49]	L	L	L	L	L	L	L
Frey et al., 2018 [50]	L	L	L	L	L	L	L
Graudins et al., 2015 [51]	L	L	L	L	L	L	L
Quinn et al., 2018 [52]	L	L	L	L	L	L	L
Reynolds et al., 2017 [53]	L	L	L	L	L	L	L
Ruffin et al., 2022 [54]	L	L	L	L	H	L	H
Younge et al., 1999 [30]	L	L	L	L	H	L	H
Adult, emergency department setting:							
Nasr Isfahani et al., 2022 [58]	L	L	L	L	L	L	L
Nazemian et al., 2019 [59]	SC	H	H	L	H	L	H

Note: H—high risk of bias; L—low risk of bias; SC—some concerns about the risk of bias.

**Table 5 jcm-12-02609-t005:** MINORS criteria quality evaluation of retrospective and prospective studies included.

Author, Year of Publication	Q1	Q2	Q3	Q4	Q5	Q6	Q7	Q8	Q9	Q10	Q11	Q12	Total
Children, prehospital emergency service:													
Murphy et al., 2017 [38]	2	2	2	2	2	2	1	0	NA	NA	NA	NA	13
Children, emergency department													
Akinsola et al., 2018 [39]	2	2	1	2	0	2	0	0	2	1	2	0	14
Anderson et al., 2022 [40]	2	2	1	2	2	2	0	0	NA	NA	NA	NA	13
Cole et al., 2009 [41]	2	2	2	2	2	2	0	2	NA	NA	NA	NA	16
Crelin et al., 2010 [42]	2	2	2	2	2	0	0	0	NA	NA	NA	NA	12
Finn et al., 2010 [43]	2	2	2	2	0	2	0	0	NA	NA	NA	NA	10
Kelly et al., 2017 [44]	2	2	0	2	2	2	0	0	2	2	1	2	17
Nemeth et al., 2017 [45]	2	2	2	2	2	2	1	0	2	0	2	2	16
Saunders et al., 2010 [46]	2	2	2	2	2	2	0	0	NA	NA	NA	NA	14
Schaefer et al., 2015 [47]	2	2	1	2	2	0	0	0	1	2	1	2	15
Adult, prehospital emergency service setting													
Tanguay et al., 2020 [55]	2	2	1	2	2	2	1	0	2	2	2	2	20
Adult, emergency department setting													
Assad et al., 2022 [56]	2	0	0	2	1	2	2	2	2	2	2	2	19
Belkouch et al., 2015 [57]	2	1	2	2	2	2	2	0	0	0	0	1	14

Note: for the following items, 0 points were given if “not reported”; 1 point if “reported but inadequate; 2 points if “reported and adequate”. NA—not admitted due to lack of comparative study. Q1—a clearly stated aim; Q2—inclusion of consecutive patients; Q3—prospective collection of data; Q4—endpoints appropriate to the aim of the study; Q5—unbiased assessment of the study endpoint; Q6—follow-up period appropriate to the aim of the study; Q7—loss to follow-up of <5%; Q8—prospective calculation of the study size. *Additional criteria in the case of comparative studies:* Q9—an adequate control group; Q10—contemporary groups; Q11—baseline equivalence of groups; Q12—adequate statistical analyses.

## Data Availability

Not Applicable.

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
