# Peer review of "Intranasal Fentanyl for Acute Pain Management in Children, Adults and Elderly Patients in the Prehospital Emergency Service and in the Emergency Department: A Systematic Review"

_jcm, 2023, doi:10.3390/jcm12072609_

Round 1
Reviewer 1 Report
Based on a review of 23 studies, this systematic review provides a comprehensive summary of the current knowledge regarding the safety and effectiveness of intranasal fentanyl (INF) for managing acute pain in various age groups (children, adults, and elderly) in prehospital emergency service (PHES) and emergency department (ED) settings. The available data are presented in a clear and insightful manner, offering a balanced overview that may be useful to a broad range of clinical workers. However, the manuscript contains numerous grammatical errors and incomplete sentences, making it difficult to follow. It requires substantial rewriting to improve its clarity and coherence.
Major comments:
1) The author stated in line 27 that there was only 1 study conducted in adult PHES, but later in line 31, they claimed that 2 PHES studies were included in the analysis of adults. This discrepancy should be addressed and clarified.
2) There are numerous grammar and syntax errors throughout the manuscript. Below are just some examples in the abstract and introduction sections:
a. Line 47, “while in ED” should be deleted.
b. line 53-55 “lack of pain recognition in fragile categories (children, elderly, patients with cognitive impairment), the underestimation of what the patient reports and the lack of communication with the patient” should all caused by “lack of training among healthcare personal on pain recognition” (Line 51). This part should be rephrased.
c. Line 47, “while in ED” should be deleted.
d. line 53-55 “lack of pain recognition in fragile categories (children, elderly, patients with cognitive impairment), the underestimation of what the patient reports and the lack of communication with the patient” should all caused by “lack of training among healthcare personal on pain recognition” (Line 51). This part should be rephrased.
e. Lines 67-71: “nasal mucosa has a large mucosal area with high blood flow and vasculature the rich capillary network of the respiratory mucosa could rapidly absorb molecules weighing less than 1000 atomic mass units transporting them to the systemic circulation without the first-pass effect” has grammar error and is hard to understand.
f. Line 71-73: “Absorption of drugs is dependent on lipophilicity and drug ionization and highly lipophilic drugs such as fentanyl can rapidly be absorbed” can be rephrased as “Absorption of drugs is dependent on lipophilicity and drug ionization. Highly lipophilic drugs such as fentanyl can rapidly be absorbed”.
g. Line 87-91: “with time to the maximal arterial concentration equal to 7 minutes with a plasma half-life of 60 minutes analgesia lasting 120-200 minutes after a single dose and only minor adverse effects related to the intranasal administration limited to mild mucosal irritation” need to be rephrased.
3) Lines 73-76: What is the logic that olfactory cell presence can help the drug reach the CSF and the brain? Olfactory cell is not a pathway for drug transport or diffusion.
Minor comments
1. What does RCT mean in the introduction?
2. Line 78: IN was not identified. It could be identified on line 63.
Author Response
Based on a review of 23 studies, this systematic review provides a comprehensive summary of the current knowledge regarding the safety and effectiveness of intranasal fentanyl (INF) for managing acute pain in various age groups (children, adults, and elderly) in prehospital emergency service (PHES) and emergency department (ED) settings. The available data are presented in a clear and insightful manner, offering a balanced overview that may be useful to a broad range of clinical workers. However, the manuscript contains numerous grammatical errors and incomplete sentences, making it difficult to follow. It requires substantial rewriting to improve its clarity and coherence.
- Reply: Dear reviewer, we really appreciate your comments on our manuscript. We are truly sorry that our work is difficult to read. However, we have thoroughly revised our manuscript to improve its readability. Moreover, according to reviewer 2, we simplified the introduction reducing its length and most of the introduction has changed. Please find attached the revised copy of the manuscript.
Major comments:
- The author stated in line 27 that there was only 1 study conducted in adult PHES, but later in line 31, they claimed that 2 PHES studies were included in the analysis of adults. This discrepancy should be addressed and clarified.
? Reply: Thank you for this advice. This is just a typing error. As shown in the tables and in the manuscript, only 1 study was conducted on adults in PHES. Accordingly, we have changed the sentence as follows: " In adults, 1 study demonstrated the efficacy of INF in the PHES setting " in lines 30-31.
2) There are numerous grammar and syntax errors throughout the manuscript. Below are just some examples in the abstract and introduction sections:
- Line 47, “while in ED” should be deleted.
? Reply: Thank you. We removed these words.
- line 53-55 “lack of pain recognition in fragile categories (children, elderly, patients with cognitive impairment), the underestimation of what the patient reports and the lack of communication with the patient” should all caused by “lack of training among healthcare personal on pain recognition” (Line 51). This part should be rephrased.
? Reply: Thank you again for your suggestion. This paragraph was rewritten as follows: “Inadequate pain management in the ED may be due to a lack of training in pain recognition and management, opioid phobia, failure to recognise pain in vulnerable groups, underestimation of patient-reported pain and poor communication with patients [12-15].”
- Line 47, “while in ED” should be deleted.
? Reply: Delated accordingly.
- line 53-55 “lack of pain recognition in fragile categories (children, elderly, patients with cognitive impairment), the underestimation of what the patient reports and the lack of communication with the patient” should all caused by “lack of training among healthcare personal on pain recognition” (Line 51). This part should be rephrased.
? Reply: rephrased as follow: “Oligoanalgesia disproportionately affects vulnerable groups such as children [8], patients with communication difficulties [9] and the elderly [10,11]. ”
- Lines 67-71: “nasal mucosa has a large mucosal area with high blood flow and vasculature the rich capillary network of the respiratory mucosa could rapidly absorb molecules weighing less than 1000 atomic mass units transporting them to the systemic circulation without the first-pass effect” has grammar error and is hard to understand.
? Reply: Again, thank you for this advice. For the sake of brevity and accordingly with reviewer 2, we removed this paragraph.
- Line 71-73: “Absorption of drugs is dependent on lipophilicity and drug ionization and highly lipophilic drugs such as fentanyl can rapidly be absorbed” can be rephrased as “Absorption of drugs is dependent on lipophilicity and drug ionization. Highly lipophilic drugs such as fentanyl can rapidly be absorbed”.
? Reply: Thank you for your precious suggestion. However, to simplify the introduction according to reviewer 2, this sentence has been extensively modified. The sentence is rewritten as follows: “IN administration of highly lipophilic drugs, such as fentanyl, is rapid, with direct entry into the CSF and brain without the hepatic first-pass effect, mak-ing it useful for analgesia, sedation, termination of seizures, reversal of narcot-ics and benzodiazepines, and treatment of hypoglycaemia [17-23]”
- Line 87-91: “with time to the maximal arterial concentration equal to 7 minutes with a plasma half-life of 60 minutes analgesia lasting 120-200 minutes after a single dose and only minor adverse effects related to the intranasal administration limited to mild mucosal irritation” need to be rephrased.
? Reply: Again, thank you for your precious advice. The sentence has been rephrased as follows: “Intranasally administered fentanyl (INF) is a well-tolerated, safe, and ef-fective route for pain management, with a bioavailability of 71%-89% [24,25], therapeutic levels reached within 2 minutes, a time to maximum arterial con-centration of 7 minutes [26,27], and a plasma half-life of 60 minutes [28]. A single dose provides analgesia lasting 120-200 minutes [27], with minor ad-verse effects limited to mild mucous membrane irritation [29]”
- Lines 73-76: What is the logic that olfactory cell presence can help the drug reach the CSF and the brain? Olfactory cell is not a pathway for drug transport or diffusion.
? Reply: Thank you for this question. The sentence could be misunderstood. We removed the part on the olfactory cell and modified the sentence as follows: “IN administration of highly lipophilic drugs such as fentanyl is rapid, with direct entry into the cerebrospinal fluid and brain, making it useful for analgesia, sedation, termination of seizures, reversal of narcotics and benzodiazepines, and treatment of hypoglycaemia
Minor comments
- What does RCT mean in the introduction?
? Reply: Thank you for this question. RCT stands for Randomized Controlled Trial and this acronym was removed from the introduction section.
- Line 78: IN was not identified. It could be identified on line 63
? Reply: Thank you. We identified it on line 63.

Reviewer 2 Report
This manuscript is a systematic review of the use of intranasal fentanyl in children and adults. The conclusion is that this method of administration is safe but additional studies are needed.
The major issue I have with the manuscript is its length and repetition. The introduction is excessively long. Most of the wording is more appropriate for the discussion (and also covered in this section).
The second point is the large number of Tables with lengthy descriptions of their data. I became lost in the results. The authors need to condense the text.
Author Response
This manuscript is a systematic review of the use of intranasal fentanyl in children and adults. The conclusion is that this method of administration is safe but additional studies are needed.
The major issue I have with the manuscript is its length and repetition. The introduction is excessively long. Most of the wording is more appropriate for the discussion (and also covered in this section).
- Reply: We would like to thank you for your comments on our manuscript. In the introduction, we have explained in great detail what we already know about intranasal administration and intranasal fentanyl. However, according to you, the introduction seems very long, so we have changed it extensively and created a shorter version that makes the whole manuscript easier to read.
The second point is the large number of Tables with lengthy descriptions of their data. I became lost in the results. The authors need to condense the text.
- Reply: Thank you for your feedback on our manuscript. As a systematic review on the use of INF in two different settings (pre-hospital emergency service and emergency department) in three different populations, the length of tables and results could be expected. For clarity and transparency we reported all available data to our readers to extensively provide what is already known on this topic. However, according to your consideration, some data are repeated in tables and text and both tables and text could be simplified. So, we decided to present table 4 to table 8 as appendix, and remove the paragraphs on the general characteristics of the included studies.

Round 2
Reviewer 1 Report
While the revision was significantly improved, some errors in other sections still exist. Please check the whole paper carefully. Here are a few examples:
1. Page 6, 1st paragraph in section 3.2, citation needs to be revised. The same problem also occurs in section 3.3.2.
2. 3.3.3 “1 RS reported a higher proportion of patients with a clinically significant reduction in pain levels in patients treated with INF than with subcutaneous fentanyl, without differences in time to administration” needs to be rephrased.
3. 3.3.4 (original 3.2.6.4) “1 PS reported INF effective in reducing pain after administration at each time point after administration (5, 30, 45 and 60 minutes)”, “after administration” was repeated.
4. 3.3.4 “Accounting for secondary outcomes of efficacy, one RS reported no difference in time to first rescue medication, with higher milligrams of morphine equivalent and a lower percentage of patients discharged home from the ED” is confusion.
5. Page 15, Discussion section “Of the seven studies that examined INF due to medical causes, three out of four were in children, one out of three were in adults on pain due to vaso-occlusive crisis due to sickle cell anemia” needs to be revised.
Author Response
Dear Revisor, thank you very much for your suggestions. We have put a lot of effort into improving our work, but small errors can still creep in. We thank you for your accurate and careful review. In addition, we have thoroughly reviewed all manuscripts, especially the results and discussion, and corrected several sentences for better readability and correctness.
Here a point by point response to your review:
While the revision was significantly improved, some errors in other sections still exist. Please check the whole paper carefully. Here are a few examples:
- Page 6, 1st paragraph in section 3.2, citation needs to be revised. The same problem also occurs in section 3.3.2..
→ Reply: In section 3.2 we noted only a missing square bracket and a missing dot after at al. In section 3.3.2 we committed an error in reporting secondary outcome of efficacies, corrected as follow: “Several authors have reported different secondary efficacy outcomes: One RCT reported an increased need for additional analgesia in patients treated with 50 μg/mL INF compared with 300 μg/mL INF [48], while others RCT reported no difference between INF and 1 mg/kg IN ketamine [51, 52] and 1.5 mg/kg IN ketamine [50], and one PS [45] reported no difference from comparators."
- 3.3.3 “1 RS reported a higher proportion of patients with a clinically significant reduction in pain levels in patients treated with INF than with subcutaneous fentanyl, without differences in time to administration” needs to be rephrased.
- → Reply: Thank you for your suggestion. The sentence was rephrased as follows: “One RS reported a higher proportion of patients with clinically significant pain relief in patients treated with INF than with subcutaneous fentanyl, with no differences in time to drug administration”
- 3.3.4 (original 3.2.6.4) “1 PS reported INF effective in reducing pain after administration at each time point after administration (5, 30, 45 and 60 minutes)”, “after administration” was repeated.
- → reply: According to you, “after administration” is a repetition and the sentence was modified as follows: “One PS reported that INF effectively reduced pain at each time point after administration (5, 30, 45 and 60 minutes)”
- 3.3.4 “Accounting for secondary outcomes of efficacy, one RS reported no difference in time to first rescue medication, with higher milligrams of morphine equivalent and a lower percentage of patients discharged home from the ED” is confusion.
→ Thank you again. The sentence was rephrased as follows: “Regarding secondary efficacy outcomes, one RS reported no difference in time to first analgesic administration between INF and IV morphine. However, INF resulted in higher milligrammes of morphine equivalents and a lower percentage of patients discharged home from the ED”
- Page 15, Discussion section “Of the seven studies that examined INF due to medical causes, three out of four were in children, one out of three were in adults on pain due to vaso-occlusive crisis due to sickle cell anaemia” needs to be revised.
→ Thank you for your suggestion. The sentence was rephrased as follows: “Seven of the included studies investigated the use of INF for medical reasons: in children, three out of four studies were conducted for pain caused by vaso-occlusive crises due to sickle cell disease; in adults, two out of three studies were conducted for pain due to renal colic. ”